# Effect of Sheep’s Whey Edible Coatings with a Bioprotective Culture, Kombucha Tea or Oregano Essential Oil on Cheese Characteristics

**DOI:** 10.3390/foods13244132

**Published:** 2024-12-20

**Authors:** Carlos D. Pereira, Hanna Varytskaya, Oliwia Łydzińska, Katarzyna Szkolnicka, David Gomes, Arona Pires

**Affiliations:** 1School of Agriculture, Polytechnic University of Coimbra, 3045-601 Coimbra, Portugal; david@esac.pt; 2Research Centre for Natural Resources, Environment and Society—CERNAS, 3045-601 Coimbra, Portugal; 3Department of Toxicology, Dairy Technology and Food Storage, West Pomeranian University of Technology, Papieża Pawła VI st. No 3, 71-459 Szczecin, Polandoliwia.lydzinska@spoko.pl (O.Ł.); katarzyna.szkolnicka@zut.edu.pl (K.S.)

**Keywords:** cheese, edible coatings, bioprotective cultures, kombucha tea, oregano essential oil

## Abstract

Films and coatings based on biopolymers have been extensively studied in recent years since they have less impact on the environment, can be obtained from renewable sources, have good coating and film-forming capacity, are biodegradable and can have interesting nutritional properties. In the present study, sheep’s cheese whey powder (SCWP) was used to produce edible cheese coatings. Six types of cheese samples were produced: without coating (CON); treated with natamycin (NAT); with SCWP coating without antimicrobials (WCO); with SCWP coating with a commercial bioprotective culture (WFQ); with SCWP coating with kombucha tea (WKO); and with SCWP coating with oregano essential oil (WEO). At the end of the ripening period, all the cheeses were classified as full-fat and semihard, according to the Portuguese standard. The higher hardness and adhesiveness values of samples CON, WFQ and WKO were in line with the lower moisture in defatted cheese observed in these samples, indicating that future work should address the improvement of water vapor barrier properties of the whey-based coating. The samples treated with natamycin and with oregano essential oil presented significantly lower values for hardness. Differences were also observed on titratable acidity and a_w_, both between samples and because of ripening time. The color parameters of cheese samples also presented differences, chiefly in the rind, but the highest differences observed resulted from ripening time rather than between samples. In all cases, the counts of lactobacilli and lactococci surpassed log 7 CFU/g by the end of ripening. Regarding yeast and mold counts, the samples CON and WCO presented the highest values by the end of the ripening period (>log 4 CFU/g), while sample NAT presented the lowest value (ca. log 3 CFU/g). Samples WFQ, WKO and WEO presented values which were ca. 0.5 log cycles lower than samples CON and WCO. Hence, the use of SCWP alongside bioprotective culture, kombucha tea or oregano essential oil had a positive impact in the reduction of mold counts on cheese surfaces. Future work should also evaluate the joint use of different antimicrobials.

## 1. Introduction

Cheese contamination with pathogenic bacteria represents a major concern for producers. Furthermore, the growth of molds on cheese surfaces causes product losses and may cause the presence of mycotoxins [1,2]. These toxins are low-molecular-weight metabolites produced by fungi that appear in cheese primarily as a result of indirect contamination or directly resulting from mold growth on cheese [3,4,5]. Although some studies state that cheese is an unfavorable matrix for mycotoxin production, these metabolites are actually detected at various concentrations [1,6]. Therefore, it is important to ensure that mold contamination is controlled to maintain the safety and quality of cheese products. Nowadays, cheese manufacturers are looking for increased sustainability, functionality and microbiological safety of their products to respond to growing consumer demands. To this end, improved packaging and reduced production waste are needed. Application of films or coatings to cover cheese surfaces has long been recognized as an important treatment for the protection of the quality parameters and microbiological safety of cheeses. The most used aids to prevent mold growth in cheeses are natamycin, or pimaricin (E235), and nisin (E234). Natamycin is a secondary metabolite produced by some *Streptomyces* species, and nisin is a polycyclic peptide produced by *Lactococcus lactis*. These molecules cause significant reductions in molds without affecting the cheese’s gross composition or proteolysis. Moreover, it is reported that the addition of natamycin does not influence the counts or evolution of the bacteria related to the starter culture [7,8,9,10,11,12,13]. However, the use of E-numbered preservatives may reduce consumers’ acceptance of cheese products.

The utilization of dairy by-products on the production of edible films and coatings is generally based on the use of cheese whey. Whey-based edible coatings are biodegradable and offer a sustainable packaging solution since they provide mechanical and chemical barrier properties, microbiological stability and good sensory characteristics while being non-toxic and having low production costs [13,14,15]. These coatings are colorless, odorless and flexible and have low tensile strength and excellent oxygen permeability [15]. Their water vapor permeability is high but can be reduced by including hydrophobic components in their formulation. Thus, they can also help in maintaining moisture content, which is crucial for preserving the texture and sensory qualities of cheese [16,17,18,19,20,21]. Furthermore, whey-protein-based films and coatings have been successfully demonstrated as carriers of active ingredients such as antimicrobials, antioxidants and probiotics [22]. Incorporating antimicrobial agents in whey coatings can significantly reduce microbial contamination, extending the shelf life of cheese [16,17]. Several studies also indicate that the addition of antimicrobials to coatings does not interfere with cheeses’ sensory acceptability [16,18,19,23]. The use of bacteriocin-producing lactic acid bacteria (LAB) has been studied as a means of biopreservation in cheese, considering both public health and spoilage aspects [24]. Biopreservation is defined as using microbes, their constituents or both to control spoilage and pathogenic microorganisms. Bioprotective cultures have also been shown to inhibit the growth of specific fungi and yeasts in different types of cheese [25]. The direct addition of bioprotective cultures (BC) to the cheese-making procedure can improve the safety of cheese products, offering protection against pathogens, such as *L. monocytogenes*, and reducing mold contamination [26,27,28]. While BC have shown inhibitory action against pathogens and spoilage microorganisms, their effectiveness may vary depending on the specific strains of fungi, the application method, the types of cheese, the type of antimicrobial and the environmental conditions [29,30,31]. The inhibitory action of such cultures results from bacteriocins produced, from pH reduction and from direct competition with spoilage or pathogenic microorganisms.

Kombucha tea (KT) is a fermented tea which is obtained from a fermentative process driven by a symbiotic culture of yeast, acetic acid bacteria and LAB. The production of kombucha generates a fermented beverage and a biofilm composed of bacterial cellulose (SCOBY). Little human-based evidence is available to prove the beneficial effect of kombucha tea on humans’ health [32]. Recently, the work of DuMez-Kornegay et al. has provided mechanistic insights into how the probiotics in KT may impact the human metabolism [33]. In vitro assays also indicated a potential anti-inflammatory activity of KT [33]. KT also offers the benefits of the tea used for its preparation, namely, its antioxidant potential [34,35]. Bacteria contained in KT belong to the genera *Acetobacter*, *Gluconobacter* and, among others, yeasts of the genus *Saccharomyces* [36,37]. However, it was observed that the yeast community was significantly different in different tea substrates, on different fermentation days and between the biofilm and the kombucha tea, indicating the influence of the substrate on the fermenting microbiota [38]. Some yeast strains have probiotic activity [39]. It is reported that *Komagataeibacter rhaeticus* is the main strain responsible for bacterial cellulose production [40,41]. The use of KT in edible coatings of cheeses relies on the inhibitory activity of its microorganisms over spoilage or pathogenic competitors based on mechanisms similar to the ones of bioprotective cultures. Ashrafi and coworkers produced an active film composed of chitosan and kombucha tea (KT). The results revealed that the incorporation of KT into chitosan films improved the water vapor permeability and enhanced the antioxidant activity of the latter [42]. Kombucha has also been successfully used as a non-conventional starter culture in the production of fresh cheese. This application has shown benefits such as reduced fermentation time and high antimicrobial activity, which could be advantageous to extend shelf life and enhance safety [43,44,45,46]. Although KT and the biofilm use in food applications have been explored by several researchers, continued research and collaborative efforts are necessary to fully unlock their potential and drive their application in the food industry [47].

Essential oils (EOs) have also long been used in food preservation. Oregano essential oil (OEO) has been widely recognized for its antimicrobial, antiviral and antifungal properties [48,49,50]. It has been shown to exhibit excellent antibacterial activity, particularly against *E. coli*, *S. aureus* and *Salmonella choleraesuis* [51,52]. The antimicrobial mechanisms of OEO are attributed to its major components, carvacrol and thymol, which exhibit antibacterial, antifungal and antioxidant activities [53,54,55,56,57,58].

Taking into account the previous information, the present study aimed at comparing the use of whey-based edible coatings containing a commercial bioprotective culture, kombucha tea and oregano EO, comparing the effect of these antimicrobials with that of natamycin and with control cheeses without any coating or with the whey-based coating without antimicrobials. To the best of our knowledge, this is the first study in which kombucha tea was incorporated in cheeses.

The conjunct application of different antimicrobials also deserves attention in future studies, since synergistic effects between them may allow for the reduction of their effective concentrations. This is particularly important in the case of EOs since their application may impart undesirable flavors, reducing consumers’ acceptance.

## 2. Materials and Methods

### 2.1. Production of Sheep Cheese Whey Powders

To produce sheep’s cheese whey powder (SCWP), 500 L of sheep’s cheese whey was subjected to ultrafiltration (UF) in a pilot plant (Proquiga Biotech, A Coruña, Spain) equipped with an organic UF membrane (3838 PVDF/polysulfone, supplied by SIVE Fluid Systems, Madrid, Spain) (effective filtration area of 7 m^2^ and 10 kDa cutoff). The process was performed at 45–50 °C, with a transmembrane pressure of 3–4 bar, aiming at a volumetric concentration factor of 20 (VCF = vol. feed/vol. retentate). The concentrate was pasteurized (75 °C, 30 s) and was further concentrated by reverse osmosis before being freeze-dried in a Lyph-Lock freeze dryer (Labconco Corporation, Kansas City, MO, USA).

### 2.2. Manufacture the Coatings with Sheep Second Cheese Whey Powders

The SCWP (97.18% ± 0.02 dry matter; 43.80% ± 0.13 protein; 37.18% ± 0.27 fat; 6.72% ± 0.31 ashes) used to prepare the coatings was dissolved in distilled water by slow stirring for 30 min at 20 °C to obtain a solution with 7% *w*/*v* (protein basis) concentration. In the case of the coatings with the bioprotective culture (BC) and with the kombucha tea (KT), only half of the required amount of water was used. Therefore, a 14% (protein basis) solution was obtained, and the remaining volume of water was replaced by the cultures, which were added after the heat treatment of the coating base.

After dissolution of the SCWP, the plasticizer (glycerol) was added (in proportions allowing for the obtainment of a 1:1 protein/plasticizer ratio). The SCWP/glycerol solution was heated to 65 °C and homogenized at 15 MPa in a Rannie™ Model Bluetop homogenizer (Copenhagen, Denmark). Then, it was heated to 95 °C and passed again trough the homogenizer valve. Finally, it was left to cool at ambient temperature.

The bioprotective culture FreshQ4™, containing *Lacticaseibacillus paracasei* and *Lacticaseibacillus rhamnosus* (Chr. Hansen, Hoersholm, Denmark), was previously prepared by adding the freeze-dried culture (0.01% *w*/*v*) to defatted sterilized milk incubated at 30 °C for 15 h. After incubation, the BC was added to the previously prepared SCWP/glycerol solution. The mixture was homogenized using the Ultra-Turrax (Ultra Turrax IKA™ T18 basic, Staufen, Germany).

The preparation of the kombucha culture involved heating water at 95 °C followed by a 5 min infusion with 3.5 g/L of green tea leaves (Lord Nelson, Lidl, Coimbra, Portugal). After removing the tea bags, 7% (*w*/*v*) sucrose was added to the infusion. The mixture was let to cool down to 30 °C, and then about 100 mL of the sourdough starter and the SCOBY from a previous fermentation was added. The preparation was maintained for 10 days at room temperature, cloth-covered and not exposed to light. Then, the solution was separated from the newly formed biomass and filtered through a cloth. Finally, the KT was added to the SCWP/glycerol solution and the mixture was homogenized using the Ultra-Turrax (Ultra Turrax IKA™ T18 basic, Staufen, Germany).

To prepare the SCWP coating with oregano essential oil (0.1% of the solution), the oil was directly added to the previously prepared SCWP/glycerol solution, and the mixture was homogenized using the Ultra-Turrax (Ultra Turrax IKA™ T18 basic, Staufen, Germany).

The solution of natamycin containing 3.2 g/L in distilled water was previously prepared, and the cheeses were submerged in it for one minute.

### 2.3. Manufacture of Model Cheeses

Six variants of experimental cheeses were produced: cheeses without coating (CON); cheeses coated with an aqueous solution of natamycin (NAT); cheeses covered with SCWP coating without additives (WCO); cheeses covered with the SCWP coating plus bioprotective culture (WFQ); cheeses covered with SCWP coating plus kombucha tea (WKO); and cheeses covered with SCWP coating plus oregano essential oil (WEO). The cheese-making process has already been described elsewhere [59]. In summary, after pasteurization of the milk and stabilizing its temperature at 29.5 ± 0.5 °C, 0.04% CaCl_2_ solution, starter culture and granulated lysozyme were added. Then, animal rennet previously diluted in water was added. Once coagulation was complete, the curd was cut into small pieces to promote drainage of whey. After half of the whey was drained, an equal amount of salted water was added to the curd, and the mixture was thoroughly stirred before final draining of the whey. The curd was placed in plastic molds before being pressed and stored in a refrigerated chamber at approximated 8–9 °C for 4 h. Afterwards, the cheeses were immersed in a brine solution for 1 h 30 min. After draining, the coatings were applied and the operation repeated on the following day. Finally, the cheeses were transferred to the ripening room, where they were maintained for 28 days at 10 ± 2 °C.

### 2.4. Physicochemical Analysis

Total dry matter of cheese products was determined by oven-drying at 105 ± 1 °C (Schutzart DIN 40050-IP20 Memmert™ oven) according to NP 3544:1987 for Cheese [60].

The ash content was gravimetrically determined by incineration of dry cheese samples at 550 °C (HD-23 Hobersal™ electric muffle furnace, Lilienthal, Germany).

Fat was determined by the Gerber method (SuperVario-N Funke Gerber™ centrifuge, Berlin, Germany) according to NP 2105:1983 [61].

For the classification of cheeses, fat in dry matter (FDM) and moisture in defatted cheese (MDC) were calculated according to NP 1598:1983 [62].

Cheese water activity (a_w_) was measured at 20 °C in a Rotronic Hygrolab (Bassersdorf, Switzerland) laboratory device equipped with an HC2-AW cell. Samples were prepared by cutting a circle of cheese (2.25 cm radius and 0.5 cm height) and placing it in a cell until a constant value of equilibrium relative humidity was attained.

The pH of all samples was determined with an HI 9025 HANNA Instruments pH meter.

The titratable acidity (TA) (% lactic acid) was determined according to AOAC 920.124 for cheese [63].

The color parameters of cheese samples (rind and paste) were determined with a colorimeter (Chroma Meter Minolta™ model CR-200B colorimeter, Tokyo, Japan) calibrated with a white standard (CR-A47: Y = 94.7; x 0.313; y 0.3204). Conditions used were as follows: illuminant C, 1 cm diameter aperture, 10° standard observer. Color coordinates were measured in the CIEL*a*b* system. Three measurements were taken for each sample. Color difference (ΔEab*) was calculated according to [64]:ΔEab* = [ΔL*^2^ + Δa*^2^ + Δb*^2^]^1/2^(1)

A Stable Micro Systems™ (Godalming, UK) texture analyzer, model TA.XT Express Enhanced, was used for the texture evaluation, and the results were calculated using Specific Expression PC software (version 6.1.11.0). A TPA-type test was run with a penetration distance of 20 mm at 2 mm/s using a cylindrical probe with a diameter of 6 mm.

### 2.5. Microbiological Analysis

The microbial counts of lactic acid bacteria (LAB) of the genera *Lactobacillus* sp. and *Lactococcus* sp. were evaluated on the 1st, 14th and 28th days of ripening. Lactobacilli were enumerated at 37 °C for 48 h on MRS agar (in anaerobiosis), and lactococci were enumerated on plates at 37 °C for 48 h on M17 agar (in aerobiosis) according to ISO 7889, IDF 117 (2003) [65]. Yeasts and molds were enumerated in plates at 25 °C on Coke Rose Bengal agar according to ISO 6611 IDF 94 (2004) [66]. Analyses were carried out in triplicate along with two controls for each medium, and results are expressed as log CFU/g of cheese product.

### 2.6. Chemicals

The chemicals used were as follows: glycerol (Panreac, Barcelona, Spain); oregano essential oil (Plena Natura, Amadora, Portugal); natamycin (Nataseen™-L, Siveele, supplied by Enzilab, Maia, Portugal); granulated lysozyme (Biostar, Toledo, Spain); CaCl_2_ solution (Betelgeaux, Valencia, Spain); sulfuric acid 90–91% (Panreac, Barcelona, Spain); isoamyl alcohol (Acros, Thermofisher Scientific, Waltham, MA, USA); NaOH (Panreac, Barcelona, Spain); MRS agar (Biokar Diagnostics, Allonne, France); M17 agar (Biokar Diagnostics, Allonne, France); Coke Rose Bengal agar (Difco, supplied by VWR, Amadora, Portugal).

### 2.7. Statistical Analysis

Prior to statistical analysis, normal distribution was evaluated using the Kolmogorov–Smirnov test. The differences among cheese samples were analyzed by two-way ANOVA and the means were compared by Tukey’s post hoc test. For all mean evaluations, a significance level of *p* < 0.05 was used. Pearson’s correlations were also analyzed to evaluate data. Statistica Software version 12 was used for the analysis of results (Stasoft Europe, Hamburg, Germany).

## 3. Results

Appendix A presents the results of the two-way ANOVA test applied to the results of the physicochemical parameters of cheese samples (Appendix A). Significative differences were observed between samples and as a result of ripening time.

The evolution of the dry matter (DM) and of the water activity (a_w_) of the different cheeses over ripening can be observed in Figure 1. Significant differences regarding DM and a_w_ were observed among samples and as a result of ripening. By the end of the ripening period, the sample containing natamycin (NAT) presented the lowest value of DM, while the sample with the coating with the bioprotective culture (WFQ) showed the highest. The samples with sheep’s cheese whey powder (SCWP) coating presented higher values of DM (on average 60.4% at the 28th day) than the control sample CON and the sample containing natamycin, which presented values of 57.7% and 55.7%, respectively. The explanation for these results may be the contribution of the coating to the solids content of the cheeses and higher loss of moisture over ripening. Accordingly, by the end of the ripening time, the samples containing SCWP also presented lower MDC contents (53–54%) than samples CON and NAT (55–57%). Differences were also observed on a_w_ values, both between samples and because of ripening time. During ripening, a_w_ decreased from ca. 0.925–0.940 to values between 0.922 and 0.928 at the end of the ripening period. The slight reduction in a_w_, which is in line with the increase in dry matter, resulted from moisture loss until the surface of cheeses reached equilibrium with the atmosphere of the ripening room.

Significant correlations were observed between dry matter and fat (0.83), with MDC (−0.67) and with adhesiveness (−0.60).

Figure 2 presents the compositional parameters of the cheese samples that, according to the Portuguese standard [62], are the basis for the classification of cheeses. By the end of the ripening period, all the cheeses produced were graded as full-fat (≥ 45 < 60% fat in dry matter) and semihard cheeses (54–63% moisture in defatted cheese). MDC presented a significant positive correlation with adhesiveness (0.50). FDM correlated positively with dry matter (0.57), adhesiveness (0.58) and cohesiveness (0.56).

The titratable acidity (TA) of cheeses showed significant differences both between products and also as a result of ripening time. It increased from ca. 0.2–0.6% lactic acid to values between 1.4 and 1.8 (Figure 3B). The variation in TA as ripening time elapsed is due to the activity of the starter LAB, which metabolizes lactose to lactate, leading to acid production. The pH values of all samples were in the range 5.2–5.0, all over the ripening period, without significant differences between products. The sample containing kombucha presented the highest TA and the lowest pH at the end of ripening. The titratable acidity presented a significant negative correlation with cohesiveness (−0.59).

Figure 4 displays the texture parameters of the cheese samples. Hardness values increased from 10 to 20 N after one week of ripening, then were maintained between 20 and 35 N until the 21st day, and finally increased to 40–60 N during the last week of ripening. At the end of the ripening time, the sample with natamycin (NAT) and the one containing oregano EO (WEO) presented significantly lower values for hardness (ca. 40 N). Sample NAT also presented the lowest value for chewiness (10 N). The higher hardness and adhesiveness values of sample CON, the sample containing kombucha tea (WKO) and the one with the bioprotective culture (WFQ) are in line with the lower MDC observed in these samples. The sample containing the edible coating without added antimicrobials (WCO) presented the highest value for chewiness. Hardness presented significant positive correlations with chewiness (0.57) and the L* parameter of the rind (0.52). Adhesiveness correlated with MDC (0.50) and with dry matter (−0.60). Chewiness correlated positively with cohesiveness (0.65) and parameter a* of the rind (0.52) and negatively with parameter L* of the paste (−0.61).

Appendix A presents the results of the two-way ANOVA of texture parameters, considering products, ripening time and the interaction of both factors. In all cases, significant differences were observed except for the interaction of both factors regarding cohesiveness.

The evolution of color parameters over ripening time is displayed in Figure 5. Lightness values (L*) of the rind decreased from ca. 85–95 on the first day to ca. 55–70 at the end of the ripening period. As expected, in the case of the paste, the reduction in lightness was not so evident. Parameter a* (green–red axis) presented a similar trend both in the rind and in the paste. Considering parameter b* (blue–yellow axis), it increased from 12 to 16–20 in the rind and to 16–18 in the paste (higher yellowness). Sample WEO presented a different pattern for the evolution of parameters L* and a* of the rind compared to all other samples (Figure 5A,C). Table 1 presents the color difference values (ΔEab*) for the same sample on different days of ripening. Values higher than 3 indicate that in normal conditions, a common observer can detect differences between samples. As expected, color differences are more marked in the rind than in the paste. The highest differences observed in the rind occurred between the first and the seventh days of ripening (ΔEab* values >15). In the paste, the differences in color observed in the first week present values lower than 2. Also, color differences were more marked after the second week of ripening. The differences between the cheese samples were less marked than the differences observed for the same sample between the first and the seventh day.

Appendix A presents the results of two-way ANOVA regarding color parameters of the rind and of the paste. Significant differences were observed between samples and as a result of ripening time. Exceptions were the L* and a* parameters of the paste, which did not present differences between products and in the interaction products × time. The b* parameter also did not present differences considering the interaction products x time. A positive correlation was observed between parameters a* and b* of the paste.

The microbiological evaluation of the different cheese samples is displayed in Figure 6. Significant differences were observed between products regarding lactobacilli counts. With respect to yeast and mold counts, significant differences were also observed both regarding products and as a result of ripening time (Appendix A). In all cases, the counts of lactobacilli and lactococci surpassed log 8 CFU/g except for sample WCO, which presented counts of lactococci of the order of log 7 CFU/g by the end of ripening. Samples CON and NAT presented average values in the range 9–9.2 log CFU/g, while the remaining samples presented lower values (8.4–9.2 log CFU/g). Regarding yeast and mold counts, samples CON and WCO presented the highest values by the end of the ripening period (>log 4 CFU/g), while the cheese sample treated with natamycin presented values of ca. log 3 CFU/g. At the end of ripening, samples WFQ, WKO and WEO presented similar values, which were 0.5 log cycles lower than samples CON and WCO. Hence, it can be considered that the use of SCWP alongside bioprotective culture, kombucha tea or oregano essential oil had a positive impact in the reduction of yeast and mold counts.

Figure 7 presents the external aspect of cheeses over ripening and the aspect of cheese paste at the end of ripening. Regarding sensory evaluation, it could be observed that all samples were well accepted (Figure 8). Nevertheless, samples CON and WEO obtained higher scores when compared to all other samples. The ranking test indicated that samples CON and WEO were clearly preferred, followed by samples WFQ, WKO and NAT, which were ranked similarly. Sample WCO was the least appreciated when compared to all other samples.

## 4. Discussion

Cheese spoilage by undesired mold growth, or the presence of pathogenic microorganisms in cheese products, has been the subject of intense research to minimize these problems. Concomitantly, the use of biopolymer-based sustainable edible coatings in cheese presents several advantages compared to the use of plastic materials. Whey-based edible coatings have been the object of several studies and, although still showing some limitations, present a non-negligeable potential for the packaging of food products, particularly cheeses [22]. Therefore, this study is in line with recent developments in this area. Furthermore, the evaluation of KT as a bioprotective culture in edible coatings is seldom reported, further justifying this research effort.

The physicochemical characteristics of the cheese samples produced show similarities compared to the ones of the same type of cheese produced in a previous work but in which the coatings were produced with a mixture of sheep’s second cheese whey and whey protein isolate (2:1) [59]. In both works, the cheeses were classified similarly, according to the Portuguese standard [62]. By the end of ripening, the samples of that study presented values of dry matter in the range 60–70%, with MDC 60–63% and FDM 46–51%. The results of the moisture in defatted cheese observed in the present work (57–62%) are somewhat unexpected since the application of the coatings should have contributed to increased moisture retention. Usually, whey-based formulations include lipids in order to improve the barrier properties of the coating. Combining lipids and proteins to create a continuous, cohesive network improves the performance of films and coatings. Therefore, the addition of hydrophobic compounds to protein-based film formulations such as waxes, fats, oils and fatty acids is important in such films. However, since the SCWP had a significant amount of fat (ca. 37%), it was decided not to use additional fat in the formulation. Most probably, the characteristics of sheep’s fat are not appropriate for the purpose, since it is rich in short-chain saturated fatty acids. The values of a_w_ and the color parameters are also similar in the cheeses produced in both works. However, the pH and the TA of those cheeses are very different to the ones observed in the present work, with higher values of pH (ca. 5.4–5.7) and lower values of TA (ca. 0.1–0.2). So, it can be concluded that the activity of the starter cultures was more intense in the cheeses produced in the present work.

Comparing the data of both works, the textural parameters also present significant differences, with lower values for hardness and adhesiveness observed in the cheeses of the previous work (ca. 17–26 for hardness and ca. −25 for adhesiveness compared to 40–60 and −18 to −45, respectively). However, it is important to notice that cheese hardness determination does not depend exclusively of the cheese bulk consistency but is also influenced by the rind consistency [67]. Hence, these differences probably result from the higher thickness of the coatings produced in the present work. It is important to highlight the lower hardness and adhesiveness values of the sample containing oregano EO. Protein–lipid composite films consist of a continuous phase and a lipidic dispersed phase typically produced by emulsification. Accordingly, it can be concluded that the use of the essential oil showed a positive impact on the structure of the whey protein coating’s textural properties.

Several research works have evaluated the incorporation of antimicrobial substances in edible films and coatings [7,8,9,10,11,17,18,19,68,69]. Ramos et al. [17] report that the antimicrobial activities of the various compounds tested as part of the edible coating for cheese are dependent on the type of target microorganism (Gram-negative vs. Gram-positive bacteria, or yeasts) and the antimicrobial compound itself. Duplication of the concentration of each antimicrobial compound did not lead to any significant increase in antimicrobial activity, revealing, therefore, a poor dose effect. This observation contrasts with data reported by other authors, who observed that the antibacterial activity of whey-based films with EOs, including oregano, was concentration-dependent [70,71].

Biopreservatives such as plant-based antimicrobials and bacteriocinogenic starter cultures have also been proposed as hurdles to increase microbiological safety of cheeses, and their pathogen inhibitory properties are reported. Some studies have focused on the development of cheese coatings with entrapped bacteriocin-producing LAB [16,25,26,27,28,68,72]. The inclusion of LAB in edible coatings and films can lead to an inhibition of pathogens through competition for space and nutrients and/or through the production of antimicrobial substances [73]. Makki et. al. also evaluated the efficacy of commercial protective cultures to inhibit mold and yeast in cottage cheese. The protective cultures tested were mostly ineffective at controlling the growth of yeasts in cottage cheese. The efficacy of these protective cultures against molds was more promising, with all protective cultures showing the ability to delay visible spoilage of at least one mold strain in the study [21]. However, another report indicates that the application of coating with immobilized cells on cheeses significantly decreased the counts of yeast up to 1 log CFU/g during 14 days [16].

From the results obtained in the present work, it can be considered that sheep’s whey-based edible coatings with added protective cultures (WFQ and WKO) or oregano EO (WEO) could reduce yeast and mold counts by ca. 0.5 log cycles without affecting the activity of the starter microorganisms and the sensory properties of cheeses. The use of natamycin in the cheese coating showed the best results regarding Y&M contamination. Fajardo et al. evaluated a chitosan-based edible film as a carrier of natamycin to improve the storability of *Saloio* cheese and reported a reduction of 1.1 log CFU/g after 27 days storage [8], while Azhdari and coworkers reported 0.6 in yeasts and 0.9 log cycle reduction in mold populations by using an antimicrobial coating based on carboxymethyl cellulose and natamycin in Mozzarella cheese, doubling the shelf life of the product [9]. Hence, our results regarding the use of natamycin in cheese coatings are comparable to the above-mentioned studies.

In the cheeses produced in our previous work, the lowest value of yeast and mold counts at the end of ripening (5.6 log CFU/g) was observed for the sample containing natamycin [59]. It could be concluded that the use of neither natamycin nor the coatings based on sheep’s second cheese whey with EOs was more efficient in reducing yeast and mold counts when compared to the negative control. However, the coatings have shown improved moisture retention in cheese, while this feature was not observed in the present work.

With regard to the use of kombucha in cheese coatings, to the best of our knowledge, there are no works available in the literature. On the other hand, some works report the use of kombucha tea as a starter added to milk used for fresh cheese production [54,55,56,57]. Fresh cheeses produced with kombucha inoculum have been reported to have acceptable sensory properties, including a mild sour taste and refreshing aroma. Kombucha fresh cheese has shown favorable physicochemical properties, such as higher protein content and better textural characteristics, compared to cheeses produced with traditional starter cultures. Fresh cheese with kombucha inoculum fortified with herbal extracts such as sage and wild thyme exhibited significant antioxidant activity, suggesting that kombucha-based coatings could potentially enhance the antioxidant properties of cheese, contributing to better preservation and health benefits. It can be postulated that the advantages observed with kombucha inoculum in fresh cheese properties could translate well into cheese coatings, potentially improving the texture and nutritional profile of the coated cheese [56]. In a minced beef model, a biocomposite film based on chitosan and kombucha tea effectively served as an active packaging and extended the shelf life of minced meat as manifested in the retardation of lipid oxidation and microbial growth from 5.36 to 2.11 log CFU/g in 4 days’ storage [53]. Therefore, the use of kombucha tea in edible coatings deserves further investigation, as it has promising properties.

## 5. Conclusions

This study demonstrated that the use of sheep’s whey-based edible coatings with added antimicrobials, namely, a bioprotective culture, kombucha tea or oregano essential oils, had a positive impact in the reduction of spoilage contamination in cheeses. The cheeses with coatings containing the antimicrobials showed a reduction in yeast and mold counts of ca. 0.5 log CFU/g by the end of ripening compared to non-treated cheeses. In all cases, lactobacilli and lactococci counts exceeded log 7 CFU/g, indicating that the antimicrobials used did not have a negative impact on the starter culture. Concerning the physicochemical properties of the cheese containing SCWP coatings, it was observed that these products presented higher dry-matter values, indicating low moisture retention capacity of the coatings. Most probably, the type of lipids present in the SCWP were responsible for this observation. The higher dry matter of coated products also impacted negatively their textural properties except for the sample containing oregano EO.

The cheeses containing bioprotective culture or kombucha tea added to the coating were less appreciated by consumers, whereas the one containing oregano EO received scores similar to those of the control sample. Further studies are suggested, namely, the joint use of bioprotective cultures and essential oils in order to improve the antimicrobial activity of the coatings.

## Figures and Tables

**Figure 1 foods-13-04132-f001:**
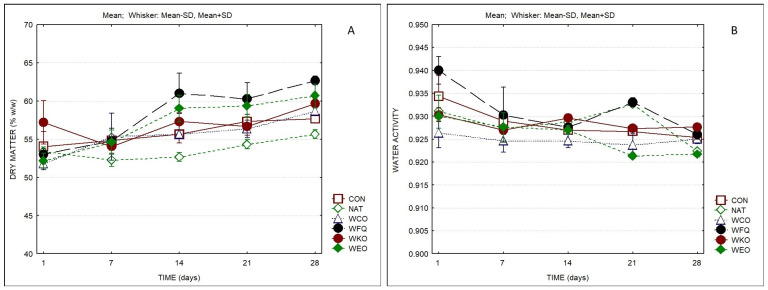
Dry matter (**A**) and water activity (**B**) of cheese samples over storage (*n* = 3). (CON) = control cheese without coating; (NAT) = cheese without coating treated with natamycin; (WCO) = cheese with SCWP coating without additives; (WFQ) = cheese with SCWP coating with bioprotective culture; (WKO) = cheese with SCWP coating with kombucha; (WEO) = cheese with SCWP coating with oregano essential oil. The same notation is used in all figures and Table 1 and Table 2.

**Figure 2 foods-13-04132-f002:**
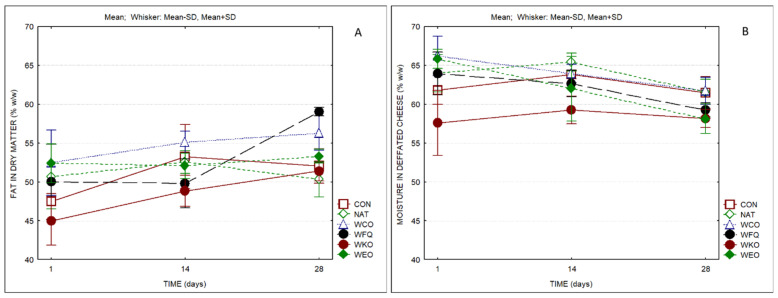
Fat in dry matter (**A**) and water in defatted cheese (**B**) of cheese samples over storage (*n* = 3).

**Figure 3 foods-13-04132-f003:**
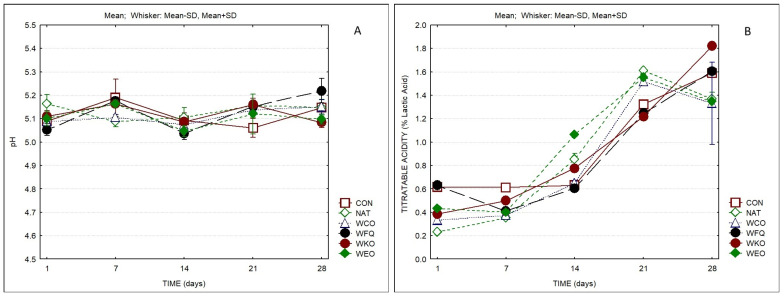
pH (**A**) and titratable acidity (**B**) of cheese samples over storage (*n* = 3).

**Figure 4 foods-13-04132-f004:**
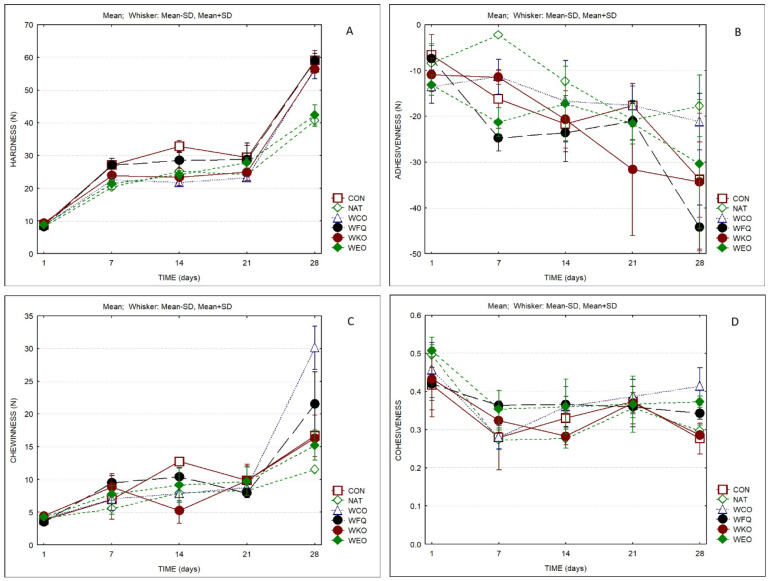
Texture parameters of cheese samples over storage (*n* = 3). (**A**) Hardness; (**B**) adhesiveness; (**C**) chewiness; (**D**) cohesiveness.

**Figure 5 foods-13-04132-f005:**
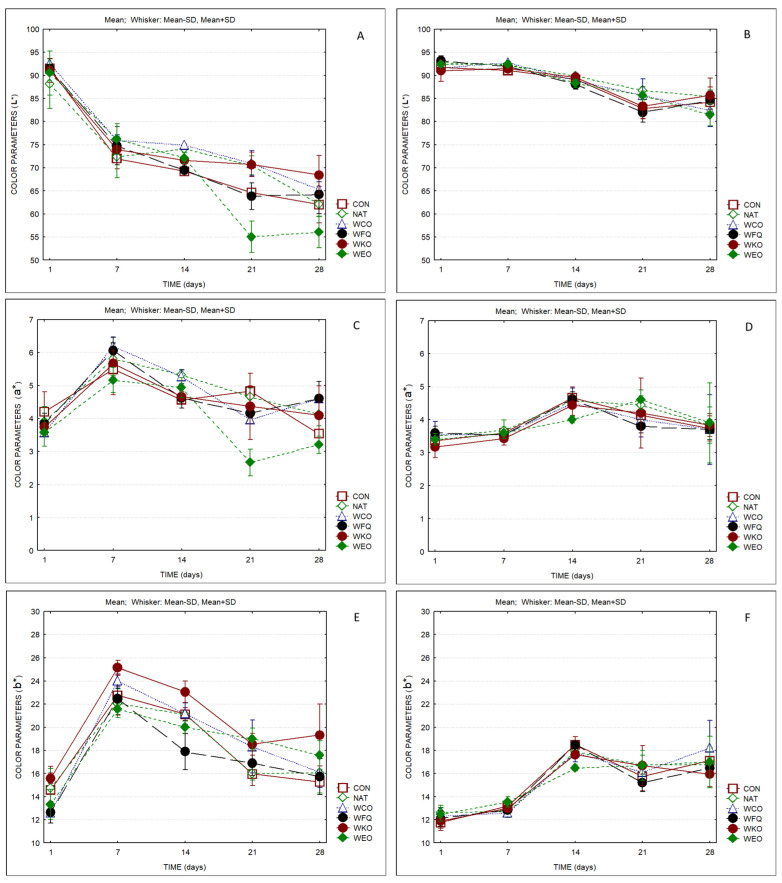
Color parameters of cheese samples over storage (*n* = 3). (**A**) L*—rind; (**B**) L*—paste; (**C**) a*—rind; (**D**) a*—paste; (**E**) b*—rind; (**F**) b*—paste.

**Figure 6 foods-13-04132-f006:**
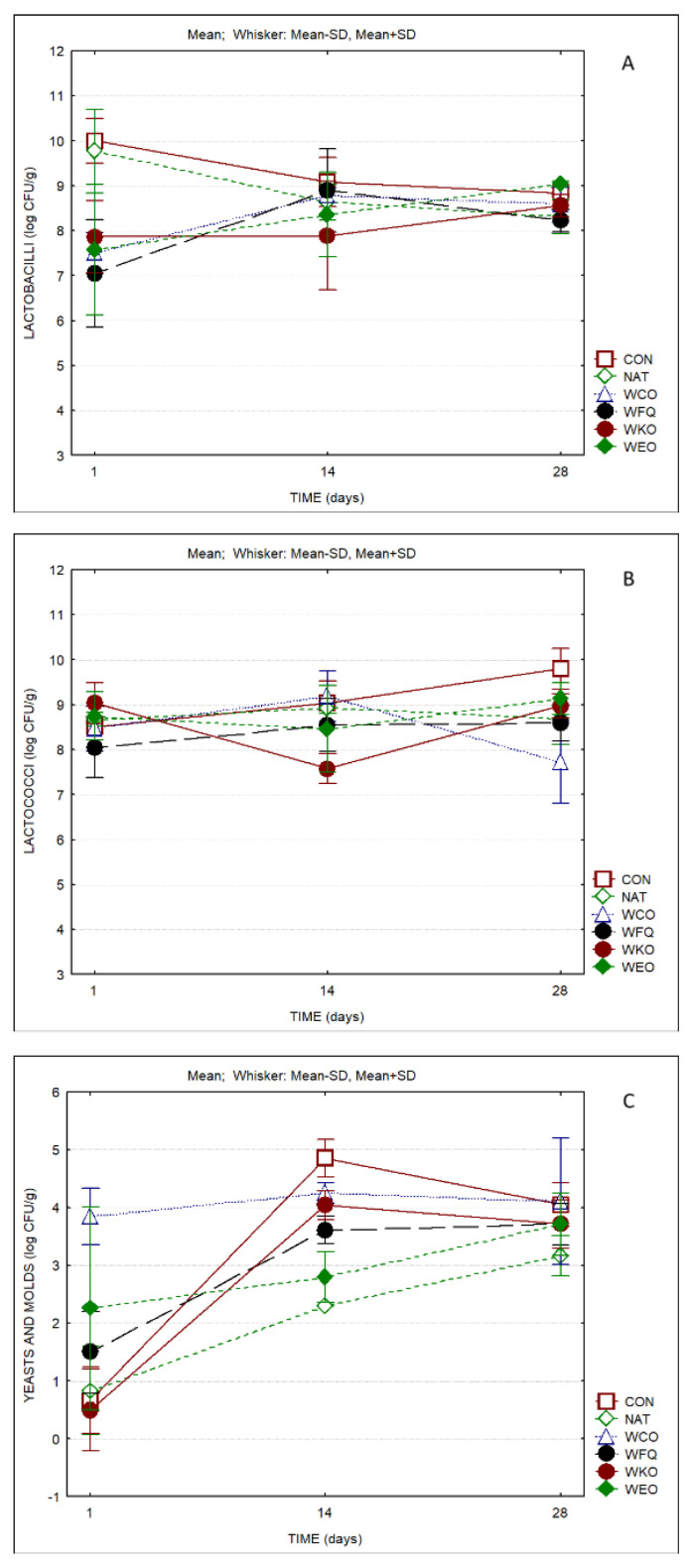
Microbial counts of cheese samples over storage (*n* = 3). (**A**) Lactobacilli; (**B**) lactococci; (**C**) yeasts and molds.

**Figure 7 foods-13-04132-f007:**
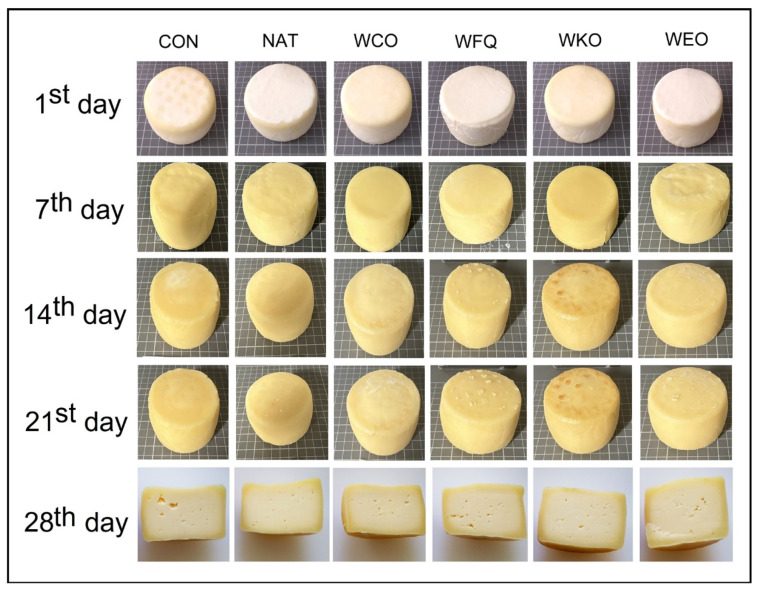
External aspect of cheese samples over storage and aspect of the cheeses’ paste on the 28th day of storage.

**Figure 8 foods-13-04132-f008:**
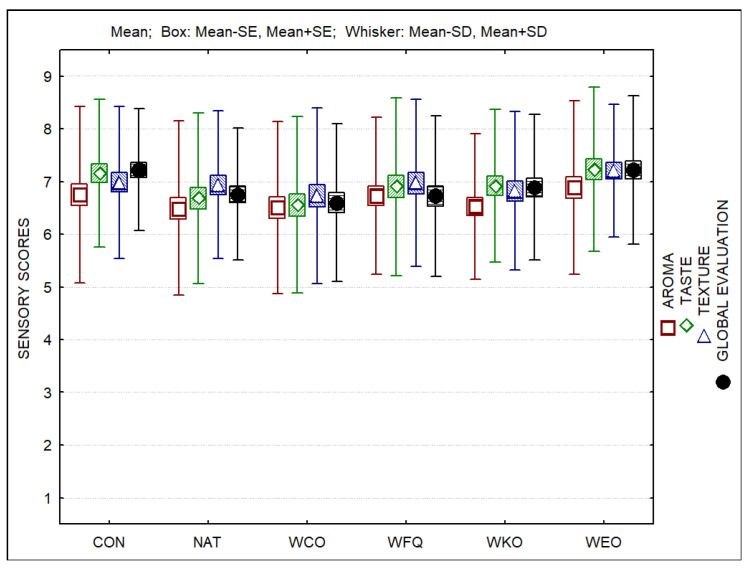
Sensory scores of cheese samples.

**Table 1 foods-13-04132-t001:** ΔEab* values of the rind and of the paste of samples on different days of ripening.

RIND	1st vs. 7th	7th vs. 14th	14th vs. 21st	21st vs. 28th
CON	21.3	3.3	7.0	3.9
NAT	17.6	2.0	7.4	8.5
WCO	20.3	3.1	8.4	2.9
WFQ	19.5	7.2	5.7	12.7
WKO	19.7	3.3	4.6	3.4
WEO	16.5	4.4	20.5	4.6
**PASTE**				
CON	1.6	5.8	6.9	2.0
NAT	0.5	5.8	3.3	1.5
WCO	1.3	24.6	17.1	10.1
WFQ	1.4	7.0	11.2	6.1
WKO	1.2	5.1	9.6	3.0
WEO	1.1	5.0	2.7	4.2

**Table 2 foods-13-04132-t002:** ΔEab* values of the rind and of the paste between samples at the end of ripening.

RIND	NAT	WCO	WFQ	WKO	WEO
CON	2.5	4.4	3.5	7.0	14.6
NAT		3.1	0.6	1.5	1.7
WCO			1.0	1.5	4.2
WFQ				17.8	5.6
WKO					12.6
**PASTE**					
CON	1.2	8.5	0.8	2.2	9.4
NAT		9.7	0.8	3.1	10.6
WCO			8.9	6.7	1.1
WFQ				2.3	6.6
WKO					0.9

## Data Availability

The original contributions presented in this study are included in the article. Further inquiries can be directed to the corresponding author.

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
