# Peer review of "Effect of Sheep’s Whey Edible Coatings with a Bioprotective Culture, Kombucha Tea or Oregano Essential Oil on Cheese Characteristics"

_foods, 2024, doi:10.3390/foods13244132_

Round 1
Reviewer 1 Report
Comments and Suggestions for Authors
Paper:
Effect of sheep’s whey edible coatings with a bioprotective culture, kombucha tea or oregano essential oil on cheese characteristics by Pereira C.D. et al., Foods.
General comments:
The present work conducted to develop a whey powder using for to produce edible cheese coatings.
I do not find the topic interesting for the food bioscience, howewer, in its present form, the manuscript presents several issus in general structure, clarity of the methods applied, description of the results, discussion and other secondary aspects that need to be modified. In general, the text was confusing to read and unclear in various parts of the text. For these reasons, kindly suggest that the manuscript be extensively revised.
Comments on the Quality of English LanguageThe article should be revised by a native speaker, because in its current form it is incomprehensible.
Author Response
Please see reply on the attached file.

Reviewer 2 Report
Comments and Suggestions for Authors
Title: Effect of sheep’s whey edible coatings with a bioprotective culture, kombucha tea or oregano essential oil on cheese characteristics
Comments:
The authors studied the effects of sheep’s whey-based edible coatings with antimicrobials (a bioprotective culture, kombucha tea or oregano essential oils) on maintaining quality of cheeses. This manuscript presents a complete study and many data are obtained. This work is meaningful for promoting the application of antimicrobial film on cheeses. However, there are serious issues with the writing of each part in the manuscript. In the current form, the innovation and focus of the manuscript have not been also emphasized. In my opinion, the manuscript requires major modifications to meet the standards required by Foods.
I give some general comments justifying the decision and that should be considered:
1. Abstract: Abstract needs to be rewritten. At present, there are a large number of results, but the logic is confusing. These results should be simplified and summarized, rather than simply listed. In addition, background introduction and research value are also necessary.
2. Introduction: Introduction needs to be rewritten. Introduction is too long and should be shortened. It is suggested to simplify the description of the antibacterials used in this study and analyze their connections and differences. The problems faced by cheese preservation and research progress on its preservation methods should be briefly summarized. The methods, design ideas, and research significance of this study should be supplemented. Innovation should be emphasized.
3. Materials and Methods: The chemical information used in this study is suggested to be displayed separately. Some methods referring to local standards are suggested to be described in detail for replication.
4. Line 147-151: Suggest transferring to Section 2.3.
5. Results: It is recommended to provide the full name when the abbreviations for the samples first appear in this section. The author presented many data. However, to emphasize the focus and accommodate more discussions, some unimportant data are suggested to be moved to the supplementary materials and briefly introduced. The format of figures is suggested to be optimized.
6. Line 283-285: This description only needs to appear in the first figure. Add a description of “The same below.”.
7. Discussion: A serious issue is that the author characterized many indicators, but only conducted superficial analysis and paid little attention to results beyond microorganisms. After reducing some unimportant data in the main text, other data should be discussed in depth, such as analysis of data trends, analysis of difference between groups, and advance compared to existing studies.
8. Conclusions: The research value and data beyond microorganisms should be presented.

Author Response
Please see reply on the attached file.

Reviewer 3 Report
Comments and Suggestions for Authors
The paper deals with the incorporation of antimicrobial substances in edible films and coatings to inhibit microbial growth in cheese. Various studies are referenced to highlight the effectiveness of different antimicrobial agents. The findings indicate that the efficacy of these antimicrobials depends on both the type of microorganism targeted and the specific compound used, with a limited dose effect observed at higher concentrations. The use of plant-based antimicrobials and bacteriocin-producing cultures is discussed, showing potential to enhance cheese safety by extending mold-free shelf life, although yeast control remains a challenge. Sheep's whey-based coatings with protective cultures or oregano essential oil were found to moderately reduce yeast and mold counts without impacting sensory qualities. Natamycin-based coatings demonstrated the highest efficacy, achieving reductions comparable to those seen in studies using chitosan or other film materials. However, a key limitation noted in the study is that the graphical kinetics data outputs in Figs 2 and 6 are based on only three ripening time points, which is insufficient for accurate modeling. Formal kinetics analysis should be applied on kinetic data observed. To ensure more reliable kinetic modeling, at least five time points are recommended. Additionally, the study did not observe improvements in moisture retention compared to previous research. While there are no current studies on the use of kombucha in cheese coatings, the paper suggests that kombucha’s positive effects on fresh cheese properties, including texture and antioxidant content, could offer promising potential for future applications in cheese preservation.
Author Response
Please see reply on the attached file.

Reviewer 4 Report
Comments and Suggestions for Authors
During the reading of the paper, no gaps or conceptual errors were detected.
Comments on the Quality of English LanguageNo comments.
Author Response
Please see reply on the attached file.

Round 2
Reviewer 1 Report
Comments and Suggestions for Authors
Paper:
Effect of sheep’s whey edible coatings with a bioprotective culture, kombucha tea or oregano essential oil on cheese characteristics by Pereira C.D. et al., Foods.
General comments:
This study was conducted to develop a whey powder using for to produce edible cheese coatings. The article has now been much improved and is more interesting and easier to understand than the previous version. The English has been much improved and the results and discussions are much clearer and more relevant.
Author Response
REV 1
Thank you for your valuable support.
REV2
Abstract: Please accurately describe the groups and the effects of each coating compared to the control.
Thank you for your valuable support. Done as proposed.
Enhance the logic and correlation of the result description, rather than cluttered stacked results.
Thank you for your valuable support. Done as proposed.
Introduction: “The problems faced by cheese preservation and research progress on its preservation methods should be briefly summarized. The methods, design ideas, and research significance of this study should be supplemented. Innovation should be emphasized.”
Please note that the questions are related to the introduction section (Line 40-161). Need to carefully modify.
Thank you for your valuable support. Done as proposed. Introduction was reformulated and research significance of the study was highlighted.
Materials and Methods: The chemical information used in this study is suggested to be displayed separately (Chemicals).
Thank you for your valuable support. Done as proposed
REV3
Revised manuscript was vigorously rewritten. The clarity of the text was improved, however the additional grammar, style and spellcheck should be performed, to improve some ackward language terms. Correct term hidrophobic at line 63 to hydrophobic. Correct term byopolimer at line 561 to biopolymer.
Thank you for your valuable support. Manuscript was thoroughly revised and corrections made.

Reviewer 2 Report
Comments and Suggestions for Authors
Title: Effect of sheep’s whey edible coatings with a bioprotective culture, kombucha tea or oregano essential oil on cheese characteristics
Comments:
The quality of the manuscript has been obviously improved after revision. However, some issues still remain unresolved.
1. Abstract: Please accurately describe the groups and the effects of each coating compared to the control. Enhance the logic and correlation of the result description, rather than cluttered stacked results.
2. Introduction: “The problems faced by cheese preservation and research progress on its preservation methods should be briefly summarized. The methods, design ideas, and research significance of this study should be supplemented. Innovation should be emphasized.” Please note that the questions are related to the introduction section (Line 40-161). Need to carefully modify.
3. Materials and Methods: The chemical information used in this study is suggested to be displayed separately (Chemicals).
Author Response

(The authors gave the same response as above.)

Reviewer 3 Report
Comments and Suggestions for Authors
Revised manuscript was vigorously rewritten. The clarity of the text was improved, however the additional grammar, style and spellcheck should be performed, to improve some ackward language terms. Correct term hidrophobic at line 63 to hydrophobic. Correct term byopolimer at line 561 to biopolymer.
Comments on the Quality of English LanguageRevised manuscript was vigorously rewritten. The clarity of the text was improved, however the additional grammar, style and spellcheck should be performed, to improve some ackward language terms. Correct term hidrophobic at line 63 to hydrophobic. Correct term byopolimer at line 561 to biopolymer.
Author Response

(The authors gave the same response as above.)
